

# Exploration of the involvement of LncRNA in HIV-associated encephalitis using bioinformatics

Diangeng Li[1,*], Pengtao Bao[1,3,*], Zhiwei Yin[4], Lei Sun[5], Jin Feng[6], Zheng He[7], Meiling Jin[2] and Changting Liu[1]

[1] Chinese PLA General Hospital, Nanlou Respiratory Diseases Department, Beijing, China
[2] Beijing Chao-yang Hospital, Department of Nephrology, Beijing, China
[3] Department of Respiratory Medicine, The 309th Hospital of People's Liberation Army, Beijing, China
[4] Chinese PLA General Hospital, Department of Nephrology, Beijing, China
[5] Chinese PLA General Hospital, Editorial Office of Chinese Journal of Otology, Beijing, China
[6] Chinese PLA General Hospital, Department of stomotology, Beijing, China
[7] Chinese PLA General Hospital, Department of Clinical Laboratory, Beijing, China
[*] These authors contributed equally to this work.

Corresponding authors
Meiling Jin, auml_1986@hotmail.com
Changting Liu,
changtingliu1212@sohu.com

## ABSTRACT

**Background.** HIV-associated encephalitis (HIVE) is one of the common complications of HIV infection, and the pathogenesis of HIVE remains unclear, while lncRNA might be involved it. In this study, we made re-annotation on the expression profiling from the HIVE study in the public database, identified the lncRNA that might be involved in HIVE, and explored the possible mechanism.

**Methods.** In the GEO public database, the microarray expression profile (GSE35864) of three regions of brain tissues (white matter, frontal cortex and basal ganglia brain tissues) was chosen, updated annotation was performed to construct the non-cording-RNA (ncRNA) microarray data. Morpheus was used to identify the differential expressed ncRNA, and Genbank of NCBI was used to identify lncRNAs. The StarBase, PITA and miRDB databases were used to predict the target miRNAs of lncRNA. The TargetScan, PicTar and MiRanda databases were used to predict the target genes of miRNAs. The GO and KEGG pathway analysis were used to make function analysis on the targets genes.

**Results.** Seventeen differentially expressed lncRNAs were observed in the white matter of brain tissues, for which 352 target miRNAs and 6,659 target genes were predicted. The GO function analysis indicated that the lncRNAs were mainly involved in the nuclear transcription and translation processes. The KEGG pathway analysis showed that the target genes were significantly enriched in 33 signal pathways, of which 11 were clearly related to the nervous system function.

**Discussion.** The brand-new and different microarray results can be obtained through the updated annotation of the chips, and it is feasible to identify lncRNAs from ordinary chips. The results suggest that lncRNA may be involved in the occurrence and development of HIVE, which provides a new direction for further research on the diagnosis and treatment of HIVE.

## INTRODUCTION

Cognitive impairment is one of the challenges that HIV patients may face (*Clifford & Ances, 2013*). HIV-1 enters the central nervous system through the blood–brain barrier at the initial infection stage, and a virus replicating area isolated from the body is formed in the central nervous system (*Stam et al., 2013*). Before the introduction of highly active anti-retroviral therapy (HAART), many HIV patients would soon suffer severe cognitive impairment, which is called HIV associated dementia (HAD), patients with HAD usually suffer from HIV-associated encephalitis (HIVE) (*Masliah et al., 2000*). Although HAART is very effective at present, HIV-induced brain inflammation has been frequently noticed in autopsy, and neurocognitive test results are abnormal in most HIV patients (*Clifford & Ances, 2013*). Currently, HIV cannot be radically eradicated by any HIV therapy, and the anti-retroviral viruses can hardly pass the blood–brain barrier, so the central nervous system may become a virus repository that might promote the occurrence and development of HIVE (*Kumar et al., 2007*). At present, the pathogenesis of HIVE is not very clear. Studying the molecular signaling pathways invovled in HIVE would be significance for the prevention and treatment of HIVE.

In the human genome, more than 70% of the genes are transcribed into RNAs, but less than 2% of them are protein coding genes, and most of them are noncoding RNAs (*Costa, 2010*). Non-coding RNAs (ncRNAs) regulate the expression of targeted genes through various pathways, and thus participate in various life processes of cells, tissues and organisms. According to the length, ncRNAs can be divided into small ncRNAs (<200 bp) and long ncRNAs (lncRNAs) (>200 bp). In recent years, studies have found that there are interactions between RNAs of different lengths, especially the relationship between lncRNA, miRNA and mRNA, which form a regulatory network of lncRNA-miRNA-mRNA. LncRNAs could be the "molecular sponges" of miRNAs, that is, lncRNAs could bind to target miRNAs leading to the "silencing effect" attenuation of miRNAs on target genes, thereby regulating the target genes of miRNAs (*Salmena et al., 2011*). This study is aiming to identify the lncRNA might be involved in HIVE from expression profile, and explore the possible mechanisms.

In recent years, expression microarray technology has played an important role in the research on the disease occurrence and development, and many research results can be reviewed and downloaded in the public database. Because updated annotation of microarray results has always been continuing, new results and novel revelation can be accomplished through the re-annotation and analysis of published studies on microarray results. In this study, we retrieved the HIVE study related microarray data from the GEO database, re-annotated and analyzed these data, identified the lncRNAs that might be involved in the pathogenesis of HIVE, and performed the correlation analysis (the work flow was shown in Fig. 1), aiming to verify the feasibility of identifying lncRNAs from expression profile from public database and to explore the possible mechanisms of lncRNAs participating in pathogenesis of HIVE.

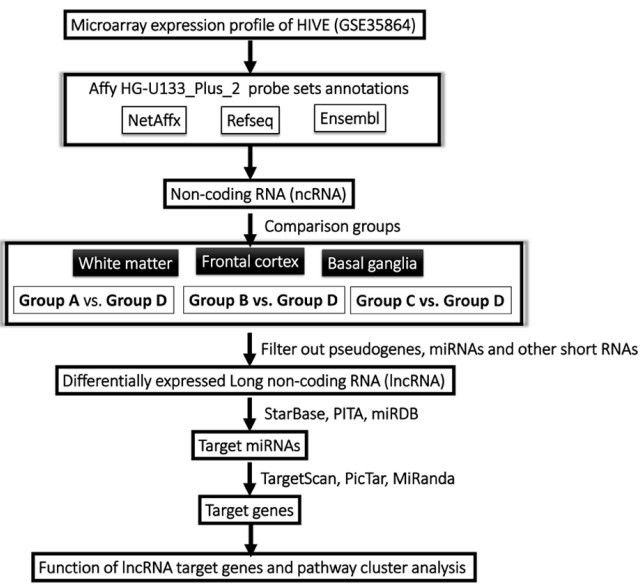

**Figure 1  Schematic overview of the work flow.**

## MATERIALS AND METHODS

### Microarray data

The Gene Expression Omnibus (GEO, http://www.ncbi.nlm.nih.gov/geo), curated by the National Center for Biotechnology Information (NCBI), is a public functional genomics database, and the data could be downloaded for free. In the GEO database, the microarray expression profile (GSE35864) in three regions of the brain, basal ganglia, white matter, and frontal cortex, in normal, HIV infected, HIV infected with neurocognitive impairment, and HIV infected with both neurocognitive impairment and encephalitis patients was chosen. Twenty-four human subjects in four groups were examined: Group A ($n = 6$) HIV-1 uninfected with no neuropathological abnormalities at autopsy; Group B ($n = 6$) HIV-1-infected (HIV+) neuropsychologically normal with no neuropathology; Group C ($n = 7$) HIV+ with substantial HIV-associated neurocognitive impairment (HAND) as defined below, with no encephalitis (HIVE) or substantial neuropathological defect; Group D ($n = 5$) HIV+ with HAND and HIVE. RNA from neocortex, white matter, and neostriatum was processed with the Affymetrix Human Genome U133 Plus 2.0 Array platform.

### Identifying differentially expressed lncRNAs

First, the latest annotation files of HG-U133_Plus_2 Annotations, CSV format, Release 36 (7/12/16) of the Affymetrix Human Genome U133 Plus 2.0 Array were downloaded from the website http://www.affymetrix.com/support/technical/annotationfilesmain.affx, including the probe set ID, gene symbol, gene title, ensemble gene ID, Refseq transcript ID and information related to the probe. The gene expression data of the chips corresponded to the probe ID, and meanwhile the probes were labeled with
Refseq transcript ID through the NetAffx annotation. The probes with the "NR_" logo were identified in Refseq ID (NR representing nonencoding RNA). Morpheus (https://software.broadinstitute.org/morpheus/) was used to analyze online and identify the differential expressed "NR_" between the groups (Group A vs. Group D, Group B vs. Group D and Group C vs. Group D) of each region tissue (white matter, frontal cortex and basal ganglia brain tissues). This analysis was based on the $t$-test and adjusted according to the characteristic that the noise of microarray data was correlated with the peak value of expression data. We considered that $p$-value <0.01 was statistically significant. The number of upregulation and downregulation was 100). The Wayne map was drawn based on the intersection of the above results to select differential "NR_" participating in the occurrence and development of HIVE. The pseudogenes, rRNAs, microRNAs and other short RNAs (including tRNAs, snRNAs and snoRNAs) were filtered out through Genbank from NCBI database. The final remainder was the differential expressed lncRNAs.

### Prediction of target miRNAs of LncRNAs and target genes

The sequences of lncRNAs were retrieved through NBCI Nucleotide (https://www.ncbi.nlm.nih.gov/nucleotide), which were then input into StarBase (http://starbase.sysu.edu.cn/), PITA (https://genie.weizmann.ac.il/pubs/mir07/mir07_data.html) and miRDB (http://www.mirdb.org/), to predict the target miRNAs of the lncRNAs. The miRNAs were input into TargetScan (http://www.targetscan.org/vert_71/), PicTar (http://pictar.mdc-berlin.de/) and MiRanda (http://www.microrna.org/microrna/home.do) to predict the corresponding target genes.

### Function of lncRNA target genes and pathway cluster analysis

The Database for Annotation, Visualization and Integrated Discovery (DAVID, http://david.abcc.ncifcrf.gov/) is an online program that provides a comprehensive set of functional annotation tools for researchers to understand biological meaning behind many genes. Gene Ontology (GO) and the Kyoto Encyclopedia of Genes and Genomes (KEGG) pathway enrichment analysis were performed for the identified DEGs using the DAVID database. After GO functional enrichment analysis, we considered the biology process terms with $p$-value <0.05 was statistically significant. For KEGG analysis, we considered the subpathway with $p$-value <0.05 was statistically significant.

## RESULTS

### Identifying differentially expressed lncRNAs

We re-annotated the GSE35864 and preliminarily retained 15,901 probes only with the "NR_" logo in the Refseq transcript ID. The Morpheus online tool was used to analyze and identify the differentially expressed probes with the "NR_" logo between the groups (Group A vs. Group D, Group B vs. Group D and Group C vs. Group D) of each region tissue (white matter, frontal cortex and basal ganglia brain tissues). As shown in Fig. 2A, there were differentially expressed probes with the "NR_" logo between different groups in the white matter, among which those that may be involved in the occurrence and development of HIVE were identified, and a total of 63 ncRNAs were identified. As

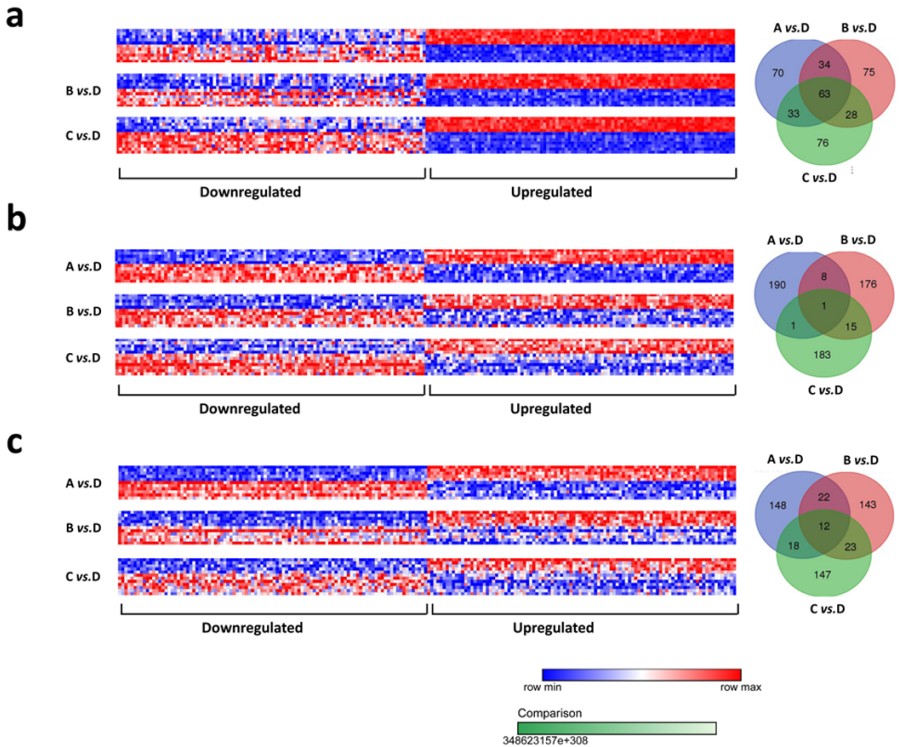

**Figure 2** **Differentially expressed ncRNAs probe sets among Group A, Group B, Group C, Group D in white matter, frontal cortex and basal ganglia.** (A) White matter; (B) frontal cortex; (C) basal ganglia. Group A: HIV-1 uninfected with no neuropathological abnormalities at autopsy; Group B: HIV 1-infected (HIV+) neuropsychologically normal with no neuropathology; Group C: HIV+ with substantial HIV-associated neurocognitive impairment (HAND) as defined below, with no encephalitis (HIVE) or substantial neuropathological defect; Group D: HIV+ with HAND and HIVE.

shown in Fig. 2B, there were differentially expressed probes with the "NR_" logo between different groups in the frontal cortex, and the intersection was selected from each group. There was only 1 probe labeled with "NR_" intersected. As shown in Fig. 2C, there were differentially expressed probes with the "NR_" logo between different groups in the basal ganglia, and the intersection was selected from each group. There were 12 common probes labeled with "NR_" intersected. All the differential expressed ncRNAs with the "NR_" logo were searched in the GenBank, and 17 lncRNAs were identified in the white matter, without any differentially expressed lncRNAs in the frontal cortex and basal ganglia. These 17 lncRNAs were LINC00308, LOC100507387, SCOC-AS1, ALMS1-IT1, LINC00639, LOC101928847, LOC100134368, ZNF670-ZNF695, SHANK2-AS3, MEG9, SNHG7, TMEM44-AS1, LRRC8C-DT, MASP1, MAPT-AS1, TBX5-AS1, LINC01770. As shown in Fig. 3, the expression of LINC00308, LOC100507387, SHANK2-AS3, SNHG7, MAPT-AS1 were significantly increasing in Group D, and the others were significantly decreasing.

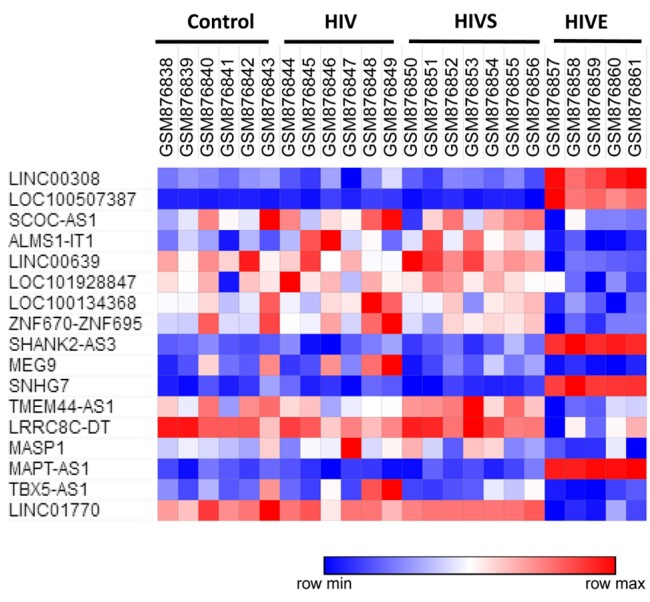

**Figure 3** Heat map of 17 differentially expressed lncRNAs among Group A, Group B, Group C, Group D in white matter. Group A: HIV-1 uninfected with no neuropathological abnormalities at autopsy; Group B: HIV-1-infected (HIV+) neuropsychologically normal with no neuropathology; Group C: HIV + with substantial HIV-associated neurocognitive impairment (HAND) as defined below, with no encephalitis (HIVE) or substantial neuropathological defect; Group D: HIV+ with HAND and HIVE.

## Prediction of target miRNAs of lncRNAs and target genes

The target micRNAs of the lncRNAs were identified from the database, and a total of 352 target miRNAs were predicted in 17 differentially expressed lncRNAs (Table 1). The 6,659 corresponding target genes of miRNAs were predicted by TargetScan, PicTar and MiRanda.

## GO cluster analysis of target genes

The 6,659 target genes were uploaded to the DAVID software, and the significant GO classification and KEGG pathway were selected. GO cell component (CC) analysis showed that the target genes were obviously clustered in the nucleus, cytoplasm, Golgi apparatus, lysosome, plasma membrane, etc. (Fig. 4A, Table 2). GO molecular function (MF) analysis revealed that the target genes were remarkably clustered in transcription factor activity, protein serine/threonine kinase activity, transcription regulator activity, ubiquitin-specific protease activity, etc. (Fig. 4B, Table 2). GO biological process (BP) analysis revealed that the target genes were significantly clustered in regulation of nucleobase, nucleoside, nucleotide and nucleic acid metabolism, signal transduction, cell communication, transport etc. (Fig. 4C, Table 2).

## KEGG pathway analysis of target genes

Table 3 showed that 33 pathways were significantly enriched with target genes ($p < 0.05$), which were obtained through KEGG analysis, including mucin type O-Glycan biosynthesis, proteoglycans in cancer, pathways in cancer, glutamatergic synapse, long-term depression

**Table 1   Differentially expressed lncRNAs in white matter and their target miRNAs.**

| No. | Probe | Official Symbol | Target miRNAs predicted |
|---|---|---|---|
| **Upregulated lncRNAs** | | | |
| 1 | NR_038400 | LINC00308 | hsa-miR-1185-2-3p; hsa-miR-1185-1-3p; hsa-miR-6833-3p; hsa-miR-4768-5p; hsa-miR-3127-5p; hsa-let-7f-2-3p; hsa-miR-5187-5p; hsa-miR-5002-3p; hsa-miR-4741; hsa-miR-4675; hsa-miR-3653-3p |
| 2 | NR_038402 /// XR_941286 /// XR_941287 /// XR_941288 /// XR_941289 /// XR_941290 /// XR_941291 /// XR_941292 /// XR_941293 | LOC100507387 | hsa-miR-6876-5p; hsa-miR-4476; hsa-miR-6865-5p; hsa-miR-6815-5p; hsa-miR-6806-5p; hsa-miR-6739-5p; hsa-miR-6733-5p; hsa-miR-3153; hsa-miR-6878-3p; hsa-miR-4779; hsa-miR-8063; hsa-miR-4738-3p; hsa-miR-5584-5p; hsa-miR-4288; hsa-miR-3688-5p; hsa-miR-1206; hsa-miR-6797-5p; hsa-miR-3978; hsa-miR-1249-5p |
| 3 | NR_073536 | SHANK2-AS3 | hsa-miR-4447; hsa-miR-4713-3p; hsa-miR-516b-5p; hsa-miR-1229-3p; hsa-miR-6832-3p; hsa-miR-504-3p; hsa-miR-1275; hsa-miR-324-5p; hsa-miR-6736-3p; hsa-miR-4443; hsa-miR-676-5p; hsa-miR-4483; hsa-miR-296-5p; hsa-miR-1224-3p; hsa-miR-96-5p; hsa-miR-1271-5p |
| 4 | NR_003672 /// NR_024542 /// NR_024543 | SNHG7 | hsa-miR-6887-5p; hsa-miR-6795-5p; hsa-miR-3201; hsa-miR-4793-5p; hsa-miR-6887-5p; hsa-miR-6795-5p; hsa-miR-3201; hsa-miR-4793-5p; hsa-miR-6836-5p; hsa-miR-6132; hsa-miR-5095; hsa-miR-4267; hsa-miR-6778-3p; hsa-miR-504-3p; hsa-miR-4262; hsa-miR-6890-5p; hsa-miR-425-5p; hsa-miR-378a-5p; hsa-miR-7151-3p |
| 5 | NR_024559 | MAPT-AS1 | hsa-miR-6772-5p; hsa-miR-4463; hsa-miR-511-5p; hsa-miR-3175; hsa-miR-532-5p; hsa-miR-4441; hsa-miR-340-5p; hsa-miR-223-5p; hsa-miR-6895-3p |
| **Downregulated lncRNAs** | | | |
| 1 | NR_033939 | SCOC-AS1 | hsa-miR-1205; hsa-miR-4508; hsa-miR-3194-5p; hsa-miR-4441; hsa-miR-6510-5p; hsa-miR-6836-5p; hsa-miR-6132; hsa-miR-4516; hsa-miR-135b-5p; hsa-miR-135a-5p; hsa-miR-3192-5p; hsa-miR-8081; hsa-miR-6876-5p; hsa-miR-4756-5p; hsa-miR-4739; hsa-miR-1321; hsa-miR-3145-5p; hsa-miR-4427; hsa-miR-4748; hsa-miR-4464; hsa-miR-651-3p; hsa-miR-4524b-3p; hsa-miR-513a-5p; hsa-miR-6794-5p; hsa-miR-4716-3p; hsa-miR-3613-3p; hsa-miR-4668-5p; hsa-miR-6867-5p; hsa-miR-6753-5p; hsa-miR-450a-1-3p; hsa-miR-4752; hsa-miR-6794-3p; hsa-miR-3188; hsa-miR-937-5p; hsa-miR-6882-3p; hsa-miR-4328; hsa-miR-4476; hsa-miR-6727-5p; hsa-miR-5003-5p; hsa-miR-4268; hsa-miR-376c-5p; hsa-miR-376b-5p; hsa-miR-1252-5p; hsa-miR-608; hsa-miR-5581-3p; hsa-miR-4651; hsa-miR-4533 |
| 2 | NR_046762 | ALMS1-IT1 | hsa-miR-4499; hsa-miR-888-5p; hsa-miR-616-5p; hsa-miR-373-5p; hsa-miR-371b-5p; hsa-miR-1285-3p; hsa-miR-539-3p; hsa-miR-485-3p; hsa-miR-3685; hsa-miR-2052; hsa-miR-3143; hsa-miR-9-3p; hsa-miR-1303; hsa-miR-548c-3p; hsa-miR-6867-3p; hsa-miR-381-3p; hsa-miR-300; hsa-miR-4801; hsa-miR-4731-3p; hsa-miR-6515-3p; hsa-miR-7844-5p; hsa-miR-506-5p; hsa-miR-5680; hsa-miR-4297; hsa-miR-155-3p; hsa-miR-660-3p; hsa-miR-6847-3p; hsa-miR-200a-3p; hsa-miR-141-3p; hsa-miR-6834-5p; hsa-miR-493-5p; hsa-miR-3911; hsa-miR-6073; hsa-miR-758-5p; hsa-miR-4426; hsa-miR-151a-3p; hsa-miR-323a-3p; hsa-miR-939-3p; hsa-miR-4262; hsa-miR-181b-5p; hsa-miR-181a-5p; hsa-miR-3653-3p; hsa-miR-7108-5p; hsa-miR-4742-5p |

**Table 1** (*continued*)

| No. | Probe | Official Symbol | Target miRNAs predicted |
|---|---|---|---|
| 3 | NR_039982 | LINC00639 | hsa-miR-6845-5p; hsa-miR-1227-5p; hsa-miR-1914-5p; hsa-miR-5088-3p; hsa-miR-6090; hsa-miR-631; hsa-miR-6762-5p; hsa-miR-4441; hsa-miR-3123; hsa-miR-6764-5p; hsa-miR-766-3p; hsa-miR-4690-5p; hsa-miR-532-3p; hsa-miR-4297; hsa-miR-4456; hsa-miR-550b-2-5p; hsa-miR-4283; hsa-miR-4281; hsa-miR-4742-5p; hsa-miR-4311; hsa-miR-330-3p; hsa-miR-8089; hsa-miR-4700-5p; hsa-miR-4667-5p |
| 4 | NR_120563 | LOC101928847 | hsa-miR-645; hsa-miR-134-3p; hsa-miR-3651; hsa-miR-5089-3p; hsa-miR-4486; hsa-miR-890; hsa-miR-6512-5p; hsa-miR-6780b-5p; hsa-miR-4725-3p; hsa-miR-4271; hsa-miR-1296-3p; hsa-miR-653-5p; hsa-miR-302f; hsa-miR-491-5p; hsa-miR-6855-5p; hsa-miR-3170; hsa-miR-1236-3p; hsa-miR-93-3p; hsa-miR-663b; hsa-miR-4769-5p; hsa-miR-4654; hsa-miR-548aw; hsa-miR-548d-5p; hsa-miR-548b-5p; hsa-miR-548ay-5p; hsa-miR-548aE−5p; hsa-miR-548ad-5p; hsa-miR-548ab; hsa-miR-548c-3p; hsa-miR-1263; hsa-miR-6071; hsa-miR-5572; hsa-miR-4665-5p; hsa-miR-1275; hsa-miR-4478; hsa-miR-518c-5p; hsa-miR-3120-3p |
| 5 | NR_024453 | LOC100134368 | hsa-miR-6742-5p; hsa-miR-663a; hsa-miR-8060; hsa-miR-1827; hsa-miR-107; hsa-miR-103a-3p; hsa-miR-4441; hsa-miR-4456; hsa-miR-6134; hsa-miR-3911; hsa-miR-4478; hsa-miR-654-5p; hsa-miR-541-3p; hsa-miR-4764-5p; hsa-miR-548x-5p; hsa-miR-548g-5p; hsa-miR-548f-5p; hsa-miR-548aj-5p; hsa-miR-4311; hsa-miR-3907; hsa-miR-4284; hsa-miR-744-5p; hsa-miR-6796-5p |
| 6 | NR_037894 | ZNF670-ZNF695 | hsa-miR-5584-5p; hsa-miR-6780a-5p; hsa-miR-4668-5p; hsa-miR-548t-3p; hsa-miR-548ap-3p; hsa-miR-548aa; hsa-miR-134-3p; hsa-miR-4306; hsa-miR-6764-5p; hsa-miR-1915-3p; hsa-miR-605-3p; hsa-miR-627-3p; hsa-miR-6779-5p; hsa-miR-3689c; hsa-miR-3689b-3p; hsa-miR-3689a-3p; hsa-miR-30b-3p; hsa-miR-1273h-5p; hsa-miR-4482-3p; hsa-miR-3128; hsa-miR-4497; hsa-miR-4297; hsa-miR-548z; hsa-miR-548h-3p; hsa-miR-548d-3p; hsa-miR-548bb-3p; hsa-miR-548ac; hsa-miR-3653-3p |
| 7 | NR_047664 | MEG9 | hsa-miR-4535; hsa-miR-4319; hsa-miR-7107-3p; hsa-miR-6753-3p; hsa-miR-4447; hsa-miR-6722-3p; hsa-miR-6069; hsa-miR-4426; hsa-miR-6721-5p; hsa-miR-7150; hsa-miR-1275; hsa-miR-7111-5p; hsa-miR-6870-5p; hsa-miR-5698; hsa-miR-4723-5p; hsa-miR-7-5p; hsa-miR-6165; hsa-miR-6502-5p; hsa-miR-1301-5p; hsa-miR-125b-5p; hsa-miR-125a-5p; hsa-miR-1226-5p; hsa-miR-4329; hsa-miR-4692; hsa-miR-4306; hsa-miR-4463; hsa-miR-4483; hsa-miR-205-3p; hsa-miR-8085; hsa-miR-6731-5p; hsa-miR-4283; hsa-miR-6794-5p; hsa-miR-4716-3p; hsa-miR-5010-5p; hsa-miR-4525; hsa-miR-4726-5p; hsa-miR-4640-5p |

**Table 1** (*continued*)

| No. | Probe | Official Symbol | Target miRNAs predicted |
|---|---|---|---|
| 8 | NR_047573 /// NR_047574 /// NR_047575 | TMEM44-AS1 | hsa-miR-4468; hsa-miR-659-3p; hsa-miR-1304-5p; hsa-miR-4478; hsa-miR-4291; hsa-miR-661; hsa-miR-6801-5p; hsa-miR-6742-3p; hsa-miR-653-3p; hsa-miR-6813-5p; hsa-miR-6085; hsa-miR-3922-5p; hsa-miR-4650-5p; hsa-miR-4717-5p; hsa-miR-3119; hsa-miR-597-3p; hsa-miR-7975; hsa-miR-296-5p; hsa-miR-2110; hsa-miR-4468; hsa-miR-659-3p; hsa-miR-1304-5p; hsa-miR-4478; hsa-miR-4291; hsa-miR-661; hsa-miR-653-3p; hsa-miR-6823-3p; hsa-miR-2114-3p; hsa-miR-6801-5p; hsa-miR-6742-3p; hsa-miR-4261; hsa-miR-6813-5p; hsa-miR-6085; hsa-miR-3922-5p; hsa-miR-4650-5p; hsa-miR-4717-5p; hsa-miR-3119; hsa-miR-597-3p; hsa-miR-7975; hsa-miR-2110; hsa-miR-4468; hsa-miR-659-3p; hsa-miR-1304-5p; hsa-miR-4478; hsa-miR-4291; hsa-miR-653-3p; hsa-miR-661; hsa-miR-6823-3p; hsa-miR-2114-3p; hsa-miR-6801-5p; hsa-miR-6742-3p; hsa-miR-4261; hsa-miR-6813-5p; hsa-miR-6085; hsa-miR-3922-5p; hsa-miR-4650-5p; hsa-miR-4717-5p; hsa-miR-3119; hsa-miR-597-3p; hsa-miR-7975 |
| 9 | NR_033981 | LRRC8C-DT | hsa-miR-3914; hsa-miR-194-3p; hsa-miR-937-3p; hsa-miR-608; hsa-miR-4651; hsa-miR-4707-5p; hsa-miR-1256; hsa-miR-561-5p; hsa-miR-4456; hsa-miR-1233-3p |
| 10 | NR_033519 | MASP1 | hsa-miR-4283; hsa-miR-6746-3p; hsa-miR-4441; hsa-miR-4763-5p; hsa-miR-1286; hsa-miR-5092; hsa-miR-4267; hsa-miR-4478; hsa-miR-1207-5p; hsa-miR-4531; hsa-miR-6768-3p; hsa-miR-4522; hsa-miR-6752-3p; hsa-miR-518c-5p; hsa-miR-6773-5p; hsa-miR-6724-5p; hsa-miR-6081; hsa-miR-6805-3p; hsa-miR-5691; hsa-miR-6718-5p; hsa-miR-203b-5p; hsa-miR-4316; hsa-miR-1976; hsa-miR-3116; hsa-miR-1254; hsa-miR-376b-3p; hsa-miR-376a-3p |
| 11 | NR_038440 | TBX5-AS1 | hsa-miR-450b-5p; hsa-miR-4455; hsa-miR-765; hsa-miR-4532; hsa-miR-5192; hsa-miR-4428; hsa-miR-4419a; hsa-miR-296-5p; hsa-miR-6859-5p; hsa-miR-3916; hsa-miR-4263; hsa-miR-6802-3p; hsa-miR-3914; hsa-miR-933; hsa-miR-6805-5p; hsa-miR-129-5p; hsa-miR-1184; hsa-miR-6828-3p; hsa-miR-6129; hsa-miR-2115-3p; hsa-miR-4463; hsa-miR-4261; hsa-miR-3116; hsa-miR-1254; hsa-miR-4516; hsa-miR-2110; hsa-miR-1273h-3p; hsa-miR-6125; hsa-miR-491-5p |
| 12 | NR_125994 /// NR_125995 /// NR_125996 | LINC01770 | hsa-miR-4723-5p; hsa-miR-7111-5p; hsa-miR-6870-5p; hsa-miR-5698; hsa-miR-1275; hsa-miR-4268; hsa-miR-4261; hsa-miR-4532; hsa-miR-604; hsa-miR-4723-5p; hsa-miR-7111-5p; hsa-miR-6870-5p; hsa-miR-5698; hsa-miR-1275; hsa-miR-4268; hsa-miR-4261; hsa-miR-4532; hsa-miR-604; hsa-miR-1227-5p; hsa-miR-4723-5p; hsa-miR-7111-5p; hsa-miR-6870-5p; hsa-miR-5698; hsa-miR-4447; hsa-miR-1275; hsa-miR-2861; hsa-miR-4268; hsa-miR-4267; hsa-miR-4261; hsa-miR-4532; hsa-miR-4707-5p; hsa-miR-6885-5p; hsa-miR-328-5p; hsa-miR-6811-3p; hsa-miR-604; hsa-miR-3620-3p |

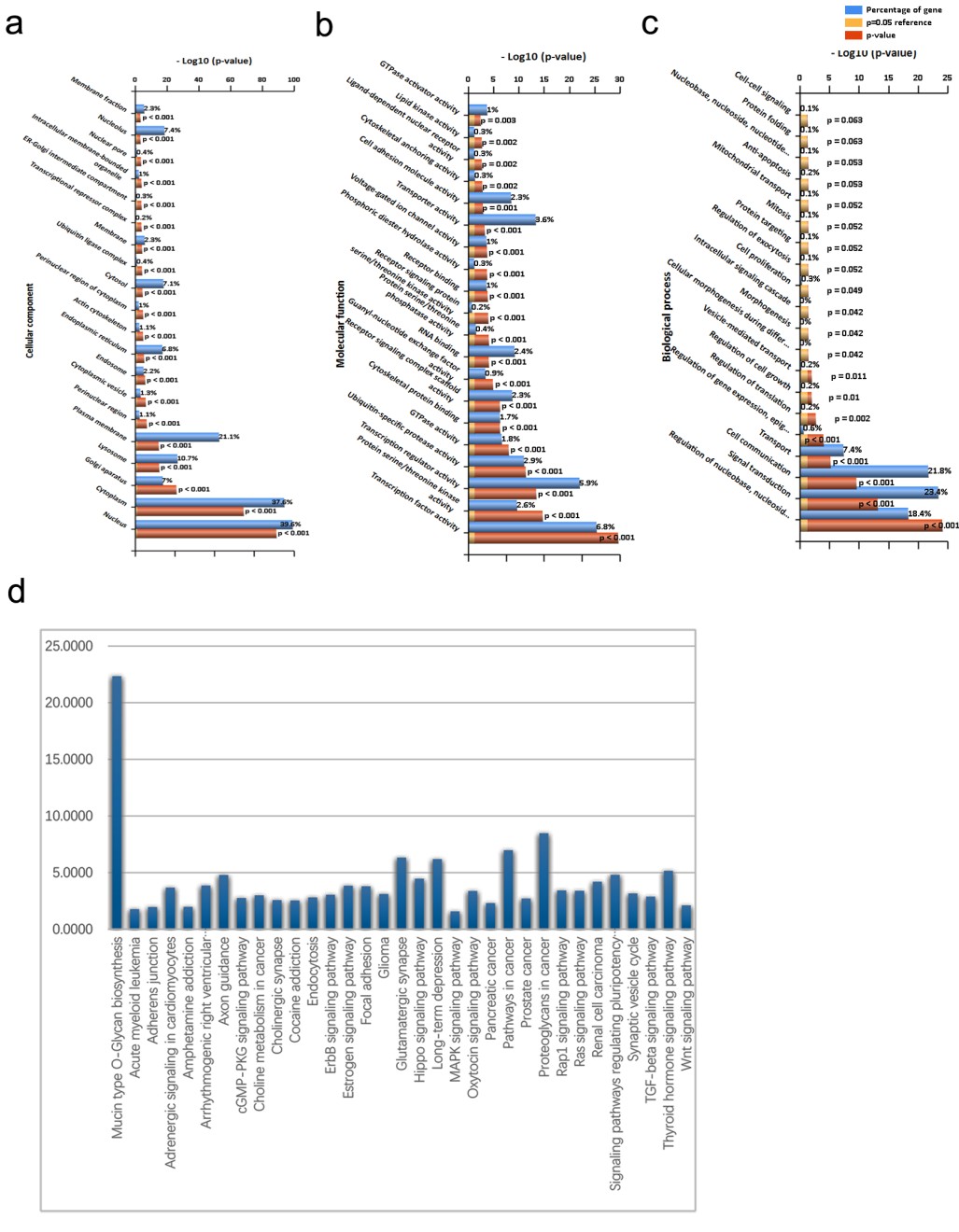

**Figure 4** **Function of lncRNA target genes and pathway cluster analysis.** (A) GO cell component (CC) analysis; (B) GO molecular function (MF) analysis; (C) GO biological process (BP) analysis; (D) KEGG pathway analysis.

**Table 2    GO analysis of targeted genes of target miRNAs of lncRNAs.**

| | No. of genes in the dataset | No. of genes in the background dataset | Percentage of genes | Fold enrichment | P-value (Hypergeometric test) | Bonferroni method | BH method | Q-value (Storey-Tibshirani method) |
|---|---|---|---|---|---|---|---|---|
| **Cellular component** | | | | | | | | |
| Nucleus | 2,639 | 5847 | 39.63058 | 1.303398 | 1.48E−89 | 1.16E−86 | 1.16E−86 | 2.80835E−86 |
| Cytoplasm | 2,501 | 5684 | 37.55819 | 1.270663 | 5.38E−69 | 4.22E−66 | 2.11E−66 | 5.1183E−66 |
| Golgi aparatus | 463 | 897 | 6.952996 | 1.490606 | 9.01E−27 | 7.07E−24 | 2.36E−24 | 5.71401E−24 |
| Lysosome | 710 | 1620 | 10.66226 | 1.265659 | 7.19E−16 | 5.64E−13 | 1.41E−13 | 3.42035E−13 |
| Plasma membrane | 1,407 | 3,479 | 21.1293 | 1.167915 | 1.7E−15 | 1.33E−12 | 2.67E−13 | 6.47199E−13 |
| Perinuclear region | 76 | 131 | 1.141313 | 1.675468 | 3.52E−08 | 2.76E−05 | 4.6E−06 | 1.11666E−05 |
| Cytoplasmic vesicle | 87 | 160 | 1.306502 | 1.570333 | 2.23E−07 | 0.000175 | 2.5E−05 | 6.05744E−05 |
| Endosome | 146 | 303 | 2.192521 | 1.391539 | 6.7E−07 | 0.000526 | 6.57E−05 | 0.000159399 |
| Endoplasmic reticulum | 453 | 1,104 | 6.802823 | 1.184963 | 3.05E−06 | 0.00239 | 0.000266 | 0.000644219 |
| Actin cytoskeleton | 70 | 132 | 1.051209 | 1.531521 | 1.04E−05 | 0.008169 | 0.000817 | 0.001981917 |
| Perinuclear region of cytoplasm | 64 | 119 | 0.961105 | 1.553224 | 1.36E−05 | 0.010642 | 0.000967 | 0.002347236 |
| Cytosol | 474 | 1,178 | 7.118186 | 1.162006 | 2.03E−05 | 0.015898 | 0.001325 | 0.003214386 |
| Ubiquitin ligase complex | 28 | 43 | 0.420484 | 1.880674 | 4.37E−05 | 0.034276 | 0.002637 | 0.006396921 |
| Membrane | 156 | 350 | 2.342694 | 1.287188 | 6.52E−05 | 0.051154 | 0.003654 | 0.008864862 |
| Transcriptional repressor complex | 14 | 17 | 0.210242 | 2.378503 | 7.49E−05 | 0.058683 | 0.003912 | 0.009491763 |
| ER-Golgi intermediate compartment | 21 | 30 | 0.315363 | 2.021761 | 8.29E−05 | 0.064958 | 0.00406 | 0.009850062 |
| Intracellular membranE−bounded organelle | 65 | 127 | 0.976123 | 1.478128 | 8.85E−05 | 0.069388 | 0.004082 | 0.009902896 |
| Nuclear pore | 26 | 42 | 0.390449 | 1.787959 | 0.000277 | 0.21717 | 0.012065 | 0.029271889 |
| Nucleolus | 492 | 1,257 | 7.388497 | 1.130329 | 0.000312 | 0.24482 | 0.012885 | 0.031262049 |
| **Molecular function** | | | | | | | | |
| Transcription factor activity | 450 | 842 | 6.757771 | 1.543387 | 2.32E−30 | 5.21E−28 | 5.21E−28 | 1.16521E−27 |
| Protein serine/threonine kinase activity | 171 | 301 | 2.567953 | 1.640629 | 1.98E−15 | 4.43E−13 | 2.22E−13 | 4.96214E−13 |
| Transcription regulator activity | 391 | 832 | 5.871753 | 1.357154 | 3.83E−14 | 8.57E−12 | 2.86E−12 | 6.39436E−12 |
| Ubiquitin-specific protease activity | 195 | 377 | 2.928368 | 1.493736 | 4.7E−12 | 1.05E−09 | 2.63E−10 | 5.88953E−10 |
| GTPase activity | 118 | 222 | 1.772038 | 1.535027 | 9.96E−09 | 2.23E−06 | 4.46E−07 | 9.98462E−07 |
| Cytoskeletal protein binding | 111 | 218 | 1.666917 | 1.470468 | 5E−07 | 0.000112 | 1.87E−05 | 4.17818E−05 |
| Receptor signaling complex scaffold activity | 154 | 322 | 2.31266 | 1.381177 | 5.95E−07 | 0.000133 | 1.9E−05 | 4.25886E−05 |

**Table 2** (*continued*)

| | No. of genes in the dataset | No. of genes in the background dataset | Percentage of genes | Fold enrichment | *P*-value (Hypergeometric test) | Bonferroni method | BH method | *Q*-value (Storey-Tibshirani method) |
|---|---|---|---|---|---|---|---|---|
| Guanyl-nucleotide exchange factor activity | 61 | 112 | 0.916053 | 1.572947 | 1.27E−05 | 0.002853 | 0.000357 | 0.000798111 |
| RNA binding | 162 | 366 | 2.432798 | 1.278259 | 7.4E−05 | 0.016579 | 0.001805 | 0.00403916 |
| Protein serine/threonine phosphatase activity | 28 | 44 | 0.420484 | 1.837941 | 8.06E−05 | 0.018046 | 0.001805 | 0.00403916 |
| Receptor signaling protein serine/threonine kinase activity | 12 | 14 | 0.180207 | 2.475567 | 0.000125 | 0.027935 | 0.00254 | 0.005683965 |
| Receptor binding | 65 | 129 | 0.976123 | 1.455213 | 0.00016 | 0.035854 | 0.002925 | 0.00654683 |
| Phosphoric diester hydrolase activity | 20 | 29 | 0.300345 | 1.991908 | 0.00017 | 0.038025 | 0.002925 | 0.00654683 |
| VoltagE−gated ion channel activity | 65 | 130 | 0.976123 | 1.44402 | 0.000213 | 0.047665 | 0.003405 | 0.007620404 |
| Transporter activity | 237 | 576 | 3.559093 | 1.188246 | 0.000563 | 0.126085 | 0.008406 | 0.018813762 |
| Cell adhesion molecule activity | 151 | 356 | 2.267608 | 1.224936 | 0.001256 | 0.281361 | 0.017585 | 0.039359191 |
| Cytoskeletal anchoring activity | 23 | 39 | 0.345397 | 1.703376 | 0.001598 | 0.357936 | 0.021055 | 0.047125868 |
| Ligand-dependent nuclear receptor activity | 21 | 35 | 0.315363 | 1.733021 | 0.001876 | 0.420189 | 0.022115 | 0.049498701 |
| Lipid kinase activity | 21 | 35 | 0.315363 | 1.733021 | 0.001876 | 0.420189 | 0.022115 | 0.049498701 |
| **Biological process** | | | | | | | | |
| Regulation of nucleobase, nucleoside, nucleotide and nucleic acid metabolism | 1,222 | 2,828 | 18.3511 | 1.247854 | 7.26E−25 | 1.29E−22 | 1.29E−22 | 3.32593E−22 |
| Signal transduction | 1,561 | 3,934 | 23.44196 | 1.145882 | 7.17E−14 | 1.28E−11 | 6.38E−12 | 1.6434E−11 |
| Cell communication | 1,449 | 3,713 | 21.76002 | 1.126977 | 2.68E−10 | 4.77E−08 | 1.59E−08 | 4.09544E−08 |
| Transport | 492 | 1,215 | 7.388497 | 1.169402 | 6.4E−06 | 0.001139 | 0.000285 | 0.000733478 |
| Regulation of gene expression, epigenetic | 38 | 66 | 0.570656 | 1.662869 | 0.000112 | 0.019988 | 0.003998 | 0.010295646 |
| Regulation of translation | 10 | 13 | 0.150173 | 2.221911 | 0.002286 | 0.406912 | 0.067819 | 0.174664437 |
| Regulation of cell growth | 13 | 21 | 0.195225 | 1.788221 | 0.009711 | 1 | 0.244311 | 0.629212828 |
| VesiclE−mediated transport | 11 | 17 | 0.16519 | 1.869187 | 0.01098 | 1 | 0.244311 | 0.629212828 |
| Cellular morphogenesis during differentiation | 3 | 3 | 0.045052 | 2.887818 | 0.041511 | 1 | 0.525479 | 1 |
| Intracellular signaling cascade | 3 | 3 | 0.045052 | 2.887818 | 0.041511 | 1 | 0.525479 | 1 |
| Morphogenesis | 3 | 3 | 0.045052 | 2.887818 | 0.041511 | 1 | 0.525479 | 1 |
| Cell proliferation | 19 | 39 | 0.285328 | 1.407265 | 0.0485 | 1 | 0.525479 | 1 |

**Table 3  KEGG pathway analysis of targeted genes of target miRNAs of lncRNAs.**

| KEGG pathway | *p*-value | Genes |
| --- | --- | --- |
| Mucin type O-Glycan biosynthesis | 4.72E−23 | C1GALT1 GALNT1 GALNTL6 GALNT3 GALNT10 GALNT13 B4GALT5 GALNT6 GALNT8 POC1B-GALNT4 GALNT7 GCNT1 GALNT4 GALNT16 |
| Proteoglycans in cancer | 3.48E−09 | CD44 PLCE1 PPP1CC ERBB4 MTOR ROCK1 VMP1 IGF1 SOS1 TGFB2 CBLB TP53 LUM RDX FZD7 CAMK2B DDX5 RAF1 SMAD2 PPP1R12B FGFR1 MET MRAS ANK2 SDC4 ITGAV ELK1 PRKCB GPC3 PIK3R1 PDPK1 TIMP3 STAT3 MAPK1 ROCK2 IGF2 PRKACA ARHGEF12 FZD3 WNT5A PTPN11 FZD6 PDCD4 ITPR1 WNT8B EGFR ANK3 CAV2 CBL CCND1 PIK3CA HPSE FRS2 ERBB3 WNT4 FAS ERBB2 GRB2 HBEGF PTK2 IQGAP1 FZD1 WNT7A CAMK2A FGF2 PRKACB PIK3CB VEGFA SOS2 FZD4 WNT9B ITGA2 PTCH1 IGF1R CAMK2D ITPR2 THBS1 GAB1 SDC2 WNT7B WNT3 EZR PPP1CB ANK1 WNT2B KRAS HSPG2 BRAF FLNA TWIST1 AKT3 DROSHA RHOA HIF1A PRKCA HGF CAMK2G WNT16 NRAS TIAM1 FN1 PPP1R12A PIK3R3 RPS6KB1 EIF4B |
| Pathways in cancer | 1.10E−07 | COL4A6 PTGER3 MTOR ROCK1 GNB5 COL4A1 SUFU CCDC6 ARHGEF11 IGF1 SOS1 TGFB2 GLI2 CBLB TP53 PTEN CDK6 PDGFRB EP300 JUP FZD7 TGFBR2 RAF1 COL4A5 SMAD2 PML EGF GNAI3 HHIP FGFR1 MET LPAR3 CREBBP MAX DVL3 PIAS2 CRK CUL2 TCF7L2 PDGFB FGFR2 ITGAV GNAI2 SHH CTNNA1 STK36 PRKCB PIK3R1 CHUK PDGFRA ADCY4 PLCB4 PLCB1 STAT3 MAPK1 ROCK2 PGF ADCY2 CTBP1 PRKACA ARHGEF12 XIAP FZD3 WNT5A EGLN3 FZD6 SMAD3 CSF2RA BIRC5 WNT8B FGF7 COL4A4 GNAI1 EGFR FGF12 IKBKB TGFBR1 CBL AXIN2 F2RL3 ITGA6 CCND1 RUNX1T1 PIK3CA NKX3-1 MSH6 BMP2 FGF9 LAMC3 WNT4 FIGF GNG13 ERBB2 TPM3 DAPK2 ARNT CDH1 NFKBIA PTK2 FZD1 PDGFA DCC WNT7A RALA GNAQ BCL2 SKP2 MLH1 TCEB1 PRKCB FGF2 STAT5B GNG2 PIK3CB CXCL12 APC BDKRB2 SMAD4 VEGFA SOS2 FZD4 GNA12 WNT9B LPAR1 ITGA2 MAPK8 EDNRA PTCH1 TPR IGF1R TRAF3 F2R ADCY5 CSF1R GNA11 BCL2L1 FGF18 RASGRP2 STK4 CASP8 MAPK10 AR HDAC2 WNT7B WNT3 FGF20 WNT2B ETS1 E2F3 KRAS CTNNA3 BRAF GNG12 APPL1 FGF1 PLD1 RALB GSK3B PPARG AKT3 RHOA HIF1A RARB PRKCA HGF LAMC1 STAT1 RASSF1 COL4A3 TGFA RUNX1 FGF14 WNT16 RB1 FGF5 NRAS MAPK9 VHL SLC2A1 FGF23 ADCY1 GNG11 CDK2 FN1 HSP90AA1 ARNT2 PIK3R3 MECOM E2F2 |
| Glutamatergic synapse | 4.88E−07 | CACNA1A HOMER1 KCNJ3 GNB5 GNG12 PRKCB GRIN2B GRM8 PLD1 ADCY4 GRIA4 CACNA1C SLC17A8 PLCB4 PLCB1 SLC38A2 MAPK1 JMJD7-PLA2G4B GRM7 GNAQ ADCY2 GLS PRKCA PRKACA GRIA1 HOMER2 GRIK4 GRM5 ADRBK1 PRKCB GNG2 GRM3 PPP3R1 ITPR1 PPP3CA GRM4 SLC17A6 GLS2 GRIA2 GRM6 GRIK3 GNAI1 SLC17A7 GRIN2A GNAI3 ADCY1 ITPR2 GNG11 ADCY5 SLC38A1 GRIA3 GRM2 PLD2 GRIK2 ADRBK2 SLC1A2 GRIN3A PLA2G4F PPP3CB DLGAP1 GNG13 TRPC1 GNAI2 GNAO1 |
| Long-term depression | 6.64E−07 | CACNA1A RAF1 NRAS GRIA2 GUCY1A2 KRAS GNA12 GNAI1 BRAF PRKCB IGF1R GNAI3 NOS1 ITPR2 PPP2R1B PLCB4 PLCB1 MAPK1 JMJD7-PLA2G4B GNAQ GRIA3 GNA11 IGF1 PPP2CA PRKCA GRIA1 PRKG1 GUCY1B3 PLA2G4F PPP2R1A GUCY1A3 ITPR1 GNAI2 GNAO1 GRID2 |
| Thyroid hormone signaling pathway | 7.16E−06 | DIO2 SIN3A PLCE1 NCOA2 KRAS MTOR PRKCB PFKFB2 PIK3R1 ATP1A4 NCOA1 GSK3B PDPK1 AKT3 PLCB4 PLCB1 MAPK1 NCOA3 MED12L HIF1A PRKCA PRKACA TBC1D4 TP53 PRKACB MED1 STAT1 MED14 PIK3CB EP300 MED17 RAF1 THRB KAT2B NRAS SLC2A1 SLC16A10 MED24 MED13 CREBBP SLC16A2 NOTCH3 CCND1 ATP2A2 PIK3CA NOTCH2 ATP1B2 MED13L PIK3R3 TSC2 ATP1B1 WNT4 HDAC2 ITGAV THRA |
| Signaling pathways regulating pluripotency of stem cells | 1.63E−05 | FZD1 ACVR2B REST HOXB1 WNT7A IGF1 FGF2 PIK3CB HNF1A APC SMAD4 FZD7 PCGF6 RAF1 FZD4 MEIS1 WNT9B SMAD2 ZIC3 IGF1R FGFR1 ACVR1 KAT6A RIF1 DVL3 JARID2 SMAD9 LIF INHBB TBX3 FGFR2 WNT7B WNT3 IL6ST WNT2B KRAS INHBA BMPR2 PCGF3 JAK2 PIK3R1 GSK3B BMPR1A AKT3 STAT3 MAPK1 ZFHX3 ID4 FZD3 PCGF5 ISL1 WNT5A FZD6 SMAD3 PAX6 WNT16 SMARCAD1 SOX2 BMI1 NRAS WNT8B POU5F1B OTX1 SMAD5 LIFR AXIN2 ACVR1C PIK3CA PIK3R3 ACVR2A KLF4 BMP2 WNT4 SKIL INHBC |

**Table 3** (*continued*)

| KEGG pathway | *p*-value | Genes |
| --- | --- | --- |
| Axon guidance | 1.65E−05 | SEMA4F EPHA5 NGEF CFL2 SEMA3A EPHA7 KRAS PTK2 ROCK1 SLIT3 EFNB1 PAK6 SEMA3E GSK3B DCC MAPK1 ROCK2 NFATC3 RHOA EPHA4 PAK7 ROBO1 EPHB6 ARHGEF12 SLIT2 ROBO2 EFNA3 CXCL12 NTN1 EPHA3 SEMA4G PPP3R1 PLXNC1 SEMA3C PPP3CA SLIT1 NRAS PAK2 SEMA5B SRGAP3 NFATC2 GNAI1 SEMA6A ABLIM1 EFNB3 PAK3 LIMK2 GNAI3 EPHA8 NTNG1 MET EFNB2 DPYSL2 UNC5B UNC5C NCK1 SEMA5A PLXNA2 UNC5D SRGAP1 NRP1 SEMA4B LRRC4 EPHB2 PPP3CB RASA1 GNAI2 |
| Hippo signaling pathway | 3.59E−05 | MOB1B PPP1CC PRKCI DLG1 CDH1 BTRC FZD1 YWHAQ SERPINE1 WNT7A LATS1 TGFB2 TEAD3 PPP2R2D GLI2 YWHAH PPP2R1A APC SMAD4 FZD7 LIMD1 TGFBR2 INADL FZD4 LATS2 WNT9B WWTR1 PPP2R2A SMAD2 WWC1 DLG2 YWHAG FRMD6 BBC3 PARD3 DVL3 CSNK1E NF2 TCF7L2 TEAD1 SNAI2 WNT7B WNT3 PPP1CB WNT2B CTNNA1 CTNNA3 BMPR2 CCND2 RASSF6 FGF1 GSK3B BMPR1A PPP2R1B CRB1 TP73 FZD3 WNT5A MPP5 RASSF1 FZD6 SMAD3 CTGF BIRC5 YAP1 WNT16 TP53BP2 SOX2 WNT8B PARD6G SAV1 GDF6 TGFBR1 AXIN2 FBXW11 DLG3 PPP2CA CCND1 BMP2 WNT4 |
| Renal cell carcinoma | 6.78E−05 | ARNT ETS1 KRAS BRAF PAK6 PIK3R1 AKT3 MAPK1 PAK7 TCEB1 SOS1 TGFB2 HIF1A HGF RAP1B PIK3CB EP300 RAP1A PTPN11 EGLN3 TGFA RAPGEF1 VEGFA SOS2 RAF1 NRAS PAK2 VHL SLC2A1 PAK3 MET CREBBP PIK3CA CRK CUL2 ARNT2 PDGFB GAB1 PIK3R3 |
| Arrhythmogenic right ventricular cardiomyopathy (ARVC) | 0.000143688 | ITGA1 CTNNA1 CTNNA3 DSG2 CACNA1C DMD LMNA PKP2 RYR2 ITGB8 DSC2 SLC8A1 CACNG3 JUP CACNA2D1 CACNB3 ITGA2 CACNB2 CACNG7 CACNG8 ITGA6 ATP2A2 ITGB6 TCF7L2 DAG1 CACNB4 GJA1 ITGAV CACNB1 CDH2 SGCD ITGA9 ITGA10 |
| Estrogen signaling pathway | 0.000148535 | FKBP5 CREB3L1 CALM3 HBEGF KRAS KCNJ3 GABBR1 CALM2 PIK3R1 KCNJ6 ADCY4 CREB3L2 AKT3 PLCB4 PLCB1 MAPK1 GNAQ ADCY2 KCNJ5 CREB3L3 SOS1 PRKACA PRKACB PIK3CB ATF2 ITPR1 SOS2 RAF1 NRAS GNAI1 EGFR GNAI3 ADCY1 SP1 ITPR2 ADCY5 GABBR2 HSP90AA1 CREB5 PIK3CA PIK3R3 CALM1 CREB1 GNAI2 GNAO1 |
| Focal adhesion | 0.000167479 | COL4A6 PPP1CC COL24A1 PTK2 ROCK1 COL5A1 PAK6 PDGFA VCL COL3A1 COL4A1 COL11A1 BCL2 IGF1 PARVA SOS1 ITGB8 PTEN PDGFRB PIK3CB TLN1 RAP1A FLT1 RAPGEF1 VEGFA VASP PARVB SOS2 COL4A5 RAF1 ITGA2 MAPK8 RELN EGF IGF1R PPP1R12B PDGFD MET THBS1 COL11A2 ITGB6 CRK PDGFB MAPK10 ITGAV COL27A1 ITGA9 PPP1CB ITGA1 TLN2 ELK1 CCND2 BRAF TNR FLNA PRKCB PIK3R1 GSK3B PDGFRA COL1A2 PDPK1 AKT3 VAV3 MAPK1 ROCK2 COL6A3 PGF RHOA PAK7 PRKCA COL6A6 MYLK4 COL1A1 XIAP HGF LAMC1 RAP1B COL4A3 ARHGAP5 PAK2 BCAR1 MAPK9 COL4A4 EGFR PAK3 CAV2 FN1 ITGA6 PPP1R12A CCND1 PIK3CA PIK3R3 LAMC3 FIGF ERBB2 ITGA10 |
| Adrenergic signaling in cardiomyocytes | 0.000215463 | TPM3 CREB3L1 CALM3 SCN7A PPP1CC PPP2R5E ATP1A4 CREB3L2 CACNA1C PPP2R5D GNAQ BCL2 CAMK2A PPP2R2D PRKACB ADRB1 PIK3CB CACNG3 PPP2R1A ATF2 CAMK2B CACNA2D1 PPP2R2A CACNG7 ATP2B2 GNAI3 KCNE1 CAMK2D ADCY5 ATP2A2 PPP2R3A CALM1 CACNB4 CACNB1 RPS6KA5 GNAI2 PPP1CB CALM2 PIK3R1 ADCY4 AKT3 PPP2R1B PLCB1 PLCB4 MAPK1 ADCY2 CREB3L3 RYR2 PRKCA PRKACA PPP1R1A SLC8A1 ATP2B4 CAMK2G ATP2B1 CACNB3 CREM CACNB2 GNAI1 ADCY1 CACNG8 TPM1 PPP2CA CREB5 PIK3CA ATP1B2 PIK3R3 ATP1B1 CREB1 |
| Rap1 signaling pathway | 0.000389106 | PFN2 CALM3 PLCE1 PRKCI CDH1 PDGFA CSF1 RALA GNAQ IGF1 FGF2 PDGFRB RAPGEF2 PIK3CB TLN1 RAP1A RAPGEF5 FLT1 INSR RAPGEF1 VEGFA VASP RAF1 NGFR LPAR1 MAGI3 EGF GRIN2A IGF1R GNAI3 PDGFD FGFR1 MET MRAS F2R LPAR3 THBS1 ADCY5 CSF1R PARD3 FGF18 KRIT1 CRK RASGRP2 PDGFB PRKD3 CALM1 CTNND1 FGFR2 GNAI2 MAP2K3 FGF20 TLN2 KRAS BRAF CALM2 PRKCB GRIN2B FGF1 PIK3R1 RALB PDGFRA ADCY4 AKT3 PLCB4 PLCB1 MAPK1 PGF ADCY2 RHOA PRKCA SPECC1L-ADORA2A HGF RAP1B EFNA3 FGF14 FGF5 RAPGEF6 NRAS BCAR1 SIPA1L3 FGF7 PARD6G GNAI1 EGFR FGF12 SIPA1L1 FGF23 TIAM1 ADCY1 F2RL3 FARP2 CNR1 MAGI2 PIK3CA ANGPT2 PIK3R3 FGF9 FIGF GNAO1 |

**Table 3** (*continued*)

| KEGG pathway | *p*-value | Genes |
|---|---|---|
| Ras signaling pathway | 0.000425038 | CALM3 PLCE1 RASA2 GNB5 PAK6 PDGFA CSF1 JMJD7-PLA2G4B RALA IGF1 RASA4 SOS1 KSR1 PLA2G12A PRKACB PLA2G2C FGF2 GNG2 PDGFRB REL PIK3CB RAP1A RAPGEF5 FLT1 INSR VEGFA SOS2 RAF1 NGFR MAPK8 EGF GRIN2A IGF1R RASGRF2 PDGFD FGFR1 MET MRAS RASAL2 CSF1R RAB5B BCL2L1 FGF18 PLD2 RASGRP2 SYNGAP1 PDGFB GAB1 STK4 PLA2G4F RASA1 MAPK10 CALM1 FGFR2 FGF20 ETS1 ELK1 KRAS ABL2 GAB2 GNG12 CALM2 PRKCB GRIN2B FGF1 PIK3R1 PLD1 RALB CHUK PDGFRA AKT3 MAPK1 PGF RHOA PAK7 PRKCA PRKACA FOXO4 HGF RAP1B RASSF1 EFNA3 PTPN11 NF1 FGF14 FGF5 KSR2 RASAL3 PAK2 NRAS FGF7 MAPK9 EGFR RAB5C FGF12 IKBKB PAK3 FGF23 TIAM1 GNG11 PIK3CA ANGPT2 PIK3R3 FGF9 FIGF GNG13 |
| Oxytocin signaling pathway | 0.000445043 | CALM3 PPP1CC KCNJ2 ROCK1 KCNJ6 CACNA1C JMJD7-PLA2G4B NFATC3 GNAQ CAMK2A PRKAA2 KCNJ5 CAMKK1 PRKACB CAMK1G PIK3CB CACNG3 GUCY1A3 CAMK2B PRKAB2 CACNA2D1 RAF1 CACNG7 GNAI3 CAMK2D PPP1R12B ITPR2 ADCY5 PLA2G4F PPP3CB CALM1 CACNB4 CACNB1 GNAI2 CAMK1 PPP1CB KRAS ELK1 KCNJ3 CALM2 PRKCB PIK3R1 ADCY4 PLCB1 PLCB4 ROCK2 MAPK1 ADCY2 RHOA RYR2 PRKCA PRKACA MYLK4 NFATC1 CAMKK2 CAMK4 PRKAA1 CAMK2G PPP3R1 ITPR1 PPP3CA CAMK1D NRAS GUCY1A2 PRKAG1 NFATC2 CACNB3 CACNB2 GNAI1 EGFR ADCY1 CD38 |
| Synaptic vesicle cycle | 0.000719318 | ATP6V1H CACNA1A SLC17A6 STX1B UNC13A CPLX3 VAMP2 UNC13C ATP6V1A ATP6V0E1 SLC17A7 CLTCL1 ATP6V0B STX2 SLC17A8 ATP6V0A1 SLC18A2 NAPA ATP6V0D2 CACNA1B ATP6V1C1 AP2M1 DNM3 ATP6V1B2 SYT1 ATP6V0D1 CLTC AP2B1 ATP6V1G2 AP2A1 ATP6V0A2 RIMS1 CPLX2 |
| Glioma | 0.000810759 | RB1 CALM3 RAF1 SOS2 NRAS E2F3 KRAS MTOR BRAF CALM2 EGFR PRKCB PDGFA EGF PIK3R1 IGF1R CAMK2D PDGFRA AKT3 MAPK1 CAMK2A IGF1 SOS1 CCND1 PRKCA PIK3CA TP53 PTEN PDGFB CDK6 PDGFRB PIK3R3 PIK3CB TGFA CALM1 CAMK2G E2F2 CAMK2B |
| ErbB signaling pathway | 0.000950519 | HBEGF ERBB4 ELK1 KRAS PTK2 ABL2 MTOR BRAF PRKCB PAK6 PIK3R1 GSK3B BTC AKT3 MAPK1 PAK7 CAMK2A SOS1 PRKCA CBLB STAT5B PIK3CB MAP2K4 TGFA CAMK2G CAMK2B SOS2 RAF1 NRAS PAK2 MAPK9 EGFR MAPK8 PAK3 EGF NRG4 CBL CAMK2D NRG3 NCK1 PIK3CA NRG1 CRK GAB1 RPS6KB1 PIK3R3 MAPK10 ERBB3 ERBB2 |
| Choline metabolism in cancer | 0.001052847 | CHPT1 DGKA DGKG KRAS MTOR PRKCB PDGFA PCYT1B PIK3R1 SLC44A5 PLD1 PDGFRA PDPK1 AKT3 MAPK1 JMJD7-PLA2G4B WASL SOS1 HIF1A TSC1 PRKCA LYPLA1 PDGFRB PIK3CB DGKI DGKH SLC44A1 SOS2 RAF1 NRAS MAPK9 GPCPD1 EGFR MAPK8 DGKB EGF WASF2 DGKZ SP1 PDGFD SLC22A3 PCYT1A PLD2 PIK3CA DGKD PIP5K1B PDGFB RPS6KB1 PIK3R3 PLA2G4F TSC2 WASF3 MAPK10 SLC44A3 SLC5A7 PIP5K1A |
| TGF-beta signaling pathway | 0.001409773 | BMPR2 INHBA ROCK1 ACVR2B BMPR1A PPP2R1B MAPK1 RHOA TGFB2 ID4 EP300 PPP2R1A SMAD3 SMAD4 TGFBR2 CHRD RBL1 SMAD2 SMAD5 GDF6 TGFBR1 SKP1 SP1 FST ACVR1 SMURF2 THBS1 ACVR1C CREBBP PPP2CA SMAD9 RPS6KB1 ZFYVE16 ACVR2A INHBB BMP2 INHBC SMURF1 |
| Endocytosis | 0.001595389 | DNAJC6 SH3GL2 ERBB4 PRKCI ASAP2 CLTCL1 AGAP1 CHMP1B HLA-F TGFB2 PSD4 CBLB AP2M1 DNM3 GIT1 ADRBK1 ADRB1 AP2A1 SH3GLB1 FLT1 TGFBR2 CYTH3 ASAP1 EHD4 ACAP2 ARAP2 SMAD2 PML EGF IQSEC3 IGF1R TFRC EPN2 MET DAB2 RUFY1 PSD2 F2R SH3KBP1 SH3GL3 CSF1R RAB5B PARD3 TSG101 RNF41 PLD2 IQSEC1 RAB11A AGAP3 VTA1 RAB11FIP2 ARFGEF1 FGFR2 SMURF1 ITCH PIP5K1A RAB31 ARF5 ARF1 GIT2 SMAP2 STAM2 PLD1 PDGFRA EEA1 VPS36 STAMBP RABEP1 CCR5 RHOA EPN3 EPS15 WWP1 AGAP2 CLTC GRK1 ARF3 SMAD3 CHMP4B RAB22A RAB11FIP4 EHD2 ARFGEF2 CHMP7 PARD6G ZFYVE20 NEDD4 EGFR RAB5C CAV2 TGFBR1 CBL SMURF2 PSD3 SH3GLB2 CHMP2B RAB11FIP1 ARFGAP3 ADRBK2 PIP5K1B PDCD6IP AP2B1 ZFYVE16 VPS37D ERBB3 ASAP3 ARRB1 |
| cGMP-PKG signaling pathway | 0.001839639 | CREB3L1 CALM3 PPP1CC PRKCE KCNMA1 ROCK1 ATP1A4 IRS2 CREB3L2 MEF2D CACNA1C NFATC3 GNAQ PDE3A ADRB1 PIK3CB ATF2 GUCY1A3 BDKRB2 INSR VASP RAF1 MEF2A ADRA2A PDE5A GNA12 EDNRA ATP2B2 GNAI3 ITPR2 IRS1 PDE2A ADCY5 GNA11 ATP2A2 PPP3CB KCNMB1 CALM1 GNAI2 PPP1CB CALM2 PIK3R1 ADCY4 AKT3 PLCB1 PLCB4 ROCK2 MAPK1 ADCY2 RHOA CREB3L3 MYLK4 NFATC1 SLC8A1 SRF PRKG1 ATP2B4 PPP3R1 ITPR1 MRVI1 PPP3CA ATP2B1 GUCY1A2 NFATC2 GNAI1 ADCY1 PPP1R12A GTF2IRD1 CREB5 PIK3CA ATP1B2 MEF2C PIK3R3 GUCY1B3 ATP1B1 CREB1 |

**Table 3** (*continued*)

| KEGG pathway | *p*-value | Genes |
| --- | --- | --- |
| Prostate cancer | 0.002067438 | CREB3L1 NFKBIA E2F3 KRAS MTOR BRAF PDGFA PIK3R1 CHUK GSK3B PDGFRA CREB3L2 PDPK1 AKT3 MAPK1 BCL2 IGF1 CREB3L3 SOS1 TP53 PTEN PDGFRB PIK3CB EP300 TGFA RB1 SOS2 RAF1 NRAS EGFR IKBKB EGF IGF1R PDGFD FGFR1 CDK2 CREBBP HSP90AA1 CREB5 CCND1 PIK3CA TCF7L2 NKX3-1 PDGFB PIK3R3 AR E2F2 FGFR2 CREB1 ERBB2 |
| Cholinergic synapse | 0.002775275 | CACNA1A CREB3L1 KCNJ2 KRAS KCNJ3 GNB5 GNG12 PRKCB PIK3R1 JAK2 CHRM1 CHRNB2 CHAT KCNJ6 ADCY4 CREB3L2 CACNA1C AKT3 PLCB4 PLCB1 MAPK1 CACNA1B GNAQ ADCY2 BCL2 CAMK2A CREB3L3 PRKCA PRKACA KCNQ3 KCNQ5 PRKACB GNG2 PIK3CB CAMK4 CAMK2G ITPR1 CAMK2B NRAS GNAI1 GNAI3 ADCY1 CAMK2D ITPR2 GNG11 CHRM3 ADCY5 GNA11 CHRM2 CREB5 CHRNA7 PIK3CA KCNQ4 PIK3R3 GNG13 GNAI2 CREB1 GNAO1 SLC5A7 |
| Cocaine addiction | 0.003037997 | CREB3L1 BDNF GRIA2 GNAI1 GRIN2B GRIN2A FOSB GNAI3 CREB3L2 PDYN SLC18A2 ADCY5 CDK5R1 CREB3L3 GRM2 CREB5 PRKACA PRKACB GRIN3A GRM3 ATF2 GNAI2 CREB1 |
| Pancreatic cancer | 0.005050774 | ARHGEF6 TGFBR2 RB1 VEGFA RAF1 E2F3 MAPK9 KRAS BRAF MAPK8 EGFR SMAD2 IK-BKB EGF PIK3R1 PLD1 RALB CHUK TGFBR1 AKT3 STAT3 MAPK1 RALA BCL2L1 TGFB2 CCND1 PIK3CA TP53 CDK6 STAT1 PIK3R3 PIK3CB MAPK10 SMAD3 TGFA E2F2 ERBB2 SMAD4 |
| Wnt signaling pathway | 0.008236698 | BTRC FZD1 CSNK2A2 WNT7A NFATC3 CAMK2A GPC4 CHD8 TP53 PRKACB PORCN EP300 TBL1X APC SMAD4 FZD7 CAMK2B TBL1XR1 VANGL2 FZD4 CSNK1A1L WNT9B MAPK8 MAP3K7 CAMK2D LRP6 CREBBP DVL3 CSNK1E FRAT1 TCF7L2 PPP3CB CSNK1A1 MAPK10 DKK2 WNT7B WNT3 WNT2B CSNK2A1 CCND2 PRKCB GSK3B SFRP1 PLCB1 PLCB4 ROCK2 DAAM1 RHOA CTBP1 PRKCA PRKACA CXXC4 NFATC1 FZD3 WNT5A FZD6 SMAD3 PPP3R1 CAMK2G PPP3CA WNT16 NLK PRICKLE2 WNT8B MAPK9 NFATC2 SKP1 AXIN2 FBXW11 DAAM2 CCND1 SIAH1 WNT4 SOST |
| Amphetamine addiction | 0.010922929 | PPP1CB CREB3L1 CALM3 PPP1CC CALM2 PRKCB GRIN2B GRIA4 CREB3L2 CACNA1C PDYN SLC18A2 CAMK2A ARC CREB3L3 PRKCA PRKACA GRIA1 PRKACB ATF2 CAMK4 CAMK2G PPP3R1 CAMK2B PPP3CA GRIA2 GRIN2A FOSB CAMK2D ADCY5 GRIA3 CREB5 GRIN3A PPP3CB CALM1 CREB1 |
| Adherens junction | 0.011434632 | YES1 PVRL3 CTNNA1 CDH1 CSNK2A1 CTNNA3 IQGAP1 FER VCL CSNK2A2 PVRL1 MAPK1 WASL RHOA PVRL4 TJP1 LMO7 EP300 SMAD3 SMAD4 INSR TGFBR2 NLK SSX2IP SORBS1 EGFR SMAD2 PTPN1 IGF1R WASF2 TGFBR1 MAP3K7 FGFR1 MET FARP2 CREBBP PARD3 TCF7L2 WASF3 SNAI2 CTNND1 ERBB2 PTPRB |
| Acute myeloid leukemia | 0.017436812 | RAF1 SOS2 NRAS KRAS MTOR BRAF IKBKB PML PIK3R1 CHUK AKT3 STAT3 MAPK1 SOS1 RUNX1T1 CCND1 PIK3CA PIM2 PIM1 TCF7L2 STAT5B RPS6KB1 PIK3R3 PIK3CB RUNX1 JUP |
| MAPK signaling pathway | 0.02786507 | CACNA1A RASA2 MAP4K4 PTPRR RAP1A ATF2 SOS2 RAF1 GNA12 CACNG7 RASGRF2 MRAS MAP3K3 RPS6KA1 CRK RASGRP2 STK4 PLA2G4F CACNB4 MAP2K3 FGF20 TAOK3 DUSP6 MAP3K1 BRAF GNG12 NTRK2 CHUK PDGFRA PRKCA PPP3R1 FGF14 PAK2 NRAS MAP4K3 EGFR ELK4 FGF12 MAP3K4 TGFBR1 CACNG8 DUSP4 |

and other relevant pathways (Fig. 4D). Eleven of these pathways may be related to the nervous system, including the glutamatergic synapse, axon guidance, Rap1 signaling pathway, Ras signaling pathway, synaptic vesicle cycle, ErbB signaling pathway, TGF-beta signaling pathway, cGMP-PKG signaling pathway, cholinergic synapse, Wnt signaling pathway and MAPK signaling pathway.

## DISCUSSION

There are two main highlights in the research. (1) The differentially expressed lncRNAs were identified through the re-annotation of published microarray results. (2) The target
miRNAs of the lncRNAs and target genes were predicted using a bioinformatics method, and GO function and KEGG pathway analyses were performed to learn about the possible mechanisms of lncRNA involved in the occurrence and development of HIVE.

In recent years, hundreds of lncRNAs have been discovered, and the changes in the expression of lncRNAs have been associated with the occurrence and development of many diseases. Plenty of evidence has shown that lncRNA is involved in the replication process of the virus (*Zhang et al., 2013*) and that lncRNA is involved in the infection process of HIV through changes in the cellular environment (*Barichievy, Naidoo & Mhlanga, 2015*). However, the role of lncRNA in the occurrence and development of HIV-related encephalitis remains unclear. The mRNA, miRNA and lncRNA that were related to the diseases were identified by microarray and bioinformatic method, which has been applied in the study of many human diseases, just the same in the study of HIV-related encephalitis. Because the annotation of microarray results has been continuously updated, some new results may be obtained by the re-annotation and re-analysis of published chips in the common database. In the GEO public database, we retrieved more comprehensive microarray results of HIVE related study (multi-group and multi-organization types), and only differential analysis of the expression of mRNA in different brain tissues (the white matter, frontal cortex and basal ganglia brain tissues) of each group was carried out. We re-annotated the microarray results and identified possible ncRNA probe results to construct the ncRNA microarray results. In addition, we then compared and analyzed the results to identiry differentially expressed ncRNAs that may be involved in the occurrence and development of HIVE. We identified 63 probes with the "NR_" logo in the white matter, one probe with the "NR_" logo in the frontal cortex, and 12 probes with the "NR_" logo in basal ganglia. All the probes with the "NR_" logo were retrieved, and it was found that only 17 probes with the "NR_" logo in the white matter were identified as lncRNAs. Among these 17 lncRNAs, expression of five were increasing in Group D, and the others were decreasing. As we found differentially expressed lncRNAs only in white matter, we speculated that cerebral white matter lesions may play an important role in the pathogenesis of HIV-associated encephalitis, which was also consistent with previous research results. The central nervous system injury affected by HIV-1 usually manifested microglial nodules comprised of multinucleated giant cells and inflammatory cells. These lesions are particularly in white matter (*Fischer-Smith et al., 2001*) Neuronal damage in HIVE is generally attributed to fully activated microglia/macrophages, especially in white matter (*Roberts, Masliah & Fox, 2004*). In addition, multinucleated giant cells and perivascular demyelination leading to white matter pallor are typical features of HIVE. In addition, these lncRNAs could be used as markers of white matter damage of HIVE. Based on the lncRNA-miRNA-mRNA mechanism,we used bioinformatics tools to predict target miRNAs and target genes of these 17 lncRNAs. GO and KEGG analysis were carried out to make the correlation cluster analysis on target genes, in order to explore the potential mechanisms of lncRNAs participating in HIVE. The GO cell component (CC) analysis results revealed that the target genes were significantly clustered in the nucleus, cytoplasm, Golgi apparatus, lysosome, plasma membrane, etc. The GO molecular function (MF) analysis showed that the target genes were significantly

clustered in the transcription factor activity, protein serine/threonine kinase activity, transcription regulator activity, ubiquitin-specific protease activity and other molecular functions. The GO biological process (BP) analysis revealed that the target genes were significantly clustered in the regulation of nucleobase, nucleoside, nucleotide and nucleic acid metabolism, signal transduction, cell communication, transport and other biological processes. Therefore, it was proven that lncRNAs may be involved in the occurrence and development of HIVE by way of their participation in the process of nuclear transcription and translation. The KEGG pathway analysis showed the target genes were significantly clustered in the mucin type O-glycan biosynthesis, proteoglycans in cancer, pathways in cancer, the glutamatergic synapse, long-term depression and other relevant pathways. In previous research, it was found that a variety of pathways were involved in neurological disorders and even in the occurrence of HIVE. The pathway of glutamatergic synapse was related with the occurrence of encephalitis, and the patients with anti-NMDAR encephalitis had a diminished function of the glutamatergic synapse (*Hughes et al., 2010*). The expression changes of the glutamatergic synapse in the brain tissues were associated with the occurrence of hepatic encephalopathy (*Montana, Verkhratsky & Parpura, 2014*). For the axon guidance pathway, it has been reported of relevant gene expression impairment of the axon guidance pathway and its downstream pathway (including MAPK pathway, calcium signaling pathway, Jak-STAT signaling pathway and VEGF signaling pathway) in the brain tissues of the patients with HIV-associated dementia, which provided new ideas for the diagnosis and treatment of HIV-associated dementia (*Zhou et al., 2012*). Both Rap1 signaling pathway and Ras signaling pathway were involved in such nervous system functions as glutamatergic synaptic transmission (*Imamura et al., 2003*), synaptic excitability (*Imamura et al., 2004*), synaptic reversibility (*Masliah et al., 2004*), etc. The abnormal expression of signaling pathway can cause encephalitis and other neuronal dysfunctions. The changes in the synaptic vesicle, especially the synaptic vesicle cycle, can cause abnormal neurotransmitter activity (*Cortes-Saladelafont et al., 2016*). For the ErbB signaling pathway, there were significant changes in the gene expression of ErbB signaling pathway in the brain tissues of the patients with HIV-associated dementia (*Shityakov, Dandekar & Forster, 2015*). The TGF-$\beta$ signaling pathway was also involved in the pathophysiology of the nervous system, which can limit inflammation and reduce neurological damage in the nervous system infection process (*Cekanaviciute et al., 2014*). The TGF-$\beta$ signaling pathway was also related with the tolerance of dendritic cells (*Esebanmen & Langridge, 2017*). Together with the STAT2 signaling pathway, the TGF-$\beta$ signaling pathway can inhibit the progression of autoimMune encephalopathy (*Xu et al., 2014*). For the cGMP-PKG signaling pathway, its abnormality had something to do with the diseases of the nervous system, and reduced kinase activity in the cGMP-PKG signaling pathway was found in rats with hepatic encephalopathy. Cognitive disorders could be relieved when the NO/sGC/cGMP/PKG signaling pathway was inhibited in diabetic rats. The iNOS-NO-cGMP signaling pathway also was involved in nervous system inflammation and myelin formation (*Raposo et al., 2014*). The activation of the Wnt signaling pathway could promote the occurrence of autoimMune encephalitis (*Schneider et al., 2016*), and the Wnt signaling pathway was involved in the immunity and tolerance of dendritic

cells (*Swafford & Manicassamy, 2015*). Moreover, plasma Dickkopf-related protein 1, the antagonist of the Wnt signaling pathway, was associated with HIV-related cognitive deficits (*Yu et al., 2017*). Therefore, KEGG analysis showed that most of the significant clustering pathways were related with the function of the nervous system. Thus, the differentially expressed lncRNAs, act as "molecular sponges", could affect the function of their target miRNAs, and thereby regulating target genes. It can be confirmed that lncRNA is indeed involved in the occurrence and development of HIV-related encephalopathy through the lncRNA-miRNA-gene mechanism. In addition, it could also be indirectly confirmed that it is feasible to construct a new chips by re-annotation and identify differentially expressed lncRNA from a expression chip.

## CONCLUSIONS

The brand-new microarray results from a different perspective can be constructed through the updated annotation of the precious and published microarray data. In this study, the lncRNA results were obtained through the re-annotation of microarray data, which provides the foundation for further research on the role of lncRNA in the occurrence and development of HIVE. From the point of view of the lncRNA-miRNA-target genes, cluster analysis was performed by various bioinformatic methods to explore the role of lncRNA. GO analysis showed that lncRNA may be involved in the occurrence and development of HIVE via its participation in the nuclear transcription and translation. The KEGG pathway analysis showed that most of the KEGG pathways with statistical significance were associated with the function of the nervous system. Therefore, we can speculate that lncRNA is indeed involved in the occurrence and development of HIVE, which is of great significance for future research on lncRNA on HIVE. It is also proved that it is feasible to identify lncRNAs from public database.

### Funding
This work was supported by the National Significant Science Foundation (2015ZX09J15102-003). The funders had no role in study design, data collection and analysis, decision to publish, or preparation of the manuscript.

### Grant Disclosures
The following grant information was disclosed by the authors:
National Significant Science Foundation: 2015ZX09J15102-003.

### Competing Interests
The authors declare there are no competing interests.

### Author Contributions

- Diangeng Li and Meiling Jin conceived and designed the experiments, performed the experiments, analyzed the data, contributed reagents/materials/analysis tools, prepared figures and/or tables, authored or reviewed drafts of the paper, approved the final draft.

- Pengtao Bao, Zhiwei Yin, Lei Sun, Jin Feng and Zheng He performed the experiments, analyzed the data.
- Changting Liu conceived and designed the experiments, authored or reviewed drafts of the paper, approved the final draft.

## Data Availability

The raw measurements are provided in Data S1.

## Supplemental Information

Supplemental information for this article can be found online at http://dx.doi.org/10.7717/peerj.5721#supplemental-information.

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
