# Peer review of "Exploration of the involvement of LncRNA in HIV-associated encephalitis using bioinformatics"

_PeerJ, doi:10.7717/peerj.5721_

## Round 0.1 · original submission · Major Revisions

Dear author,

Thank you for submitting to Peer J. Your manuscript was reviewed by three independent reviewers. As you can see, two of the reviewers have raised major concern, while a third has suggested minor revisions. You are therefore advised for Major Revisions as per reviewers comments. Please address the concerns of all 3 reviewers, in particular those of Reviewers 2 and 3.

Best Regards

Ravi Tandon

Reviewer 1 ·

Basic reporting

Indeed a good effort of using bioinformatics methods to reassign and shed light on different already existing data of HIV associated encephalopathy.


however. this work falls short in concluding any new novel targets or new biomarkers that was discovered in the study.
The work only generally lists all the pathways associated with lncRNA .The discussion also excludes the expression level of the 17 lncRNA and thus the importance of which lncRNA should precede the other.
Please in ascending or descending order the lncRNAs and their targets genes which affect HIV associated encephalopathy.
control, HIV, HIVS, HIVE datasets are some what not discussed in detail once the differential expression is found

Experimental design

no comment

Validity of the findings

The findings using webbased applications are indeed correlating with many references mentioned in the discussion however if finding the pathogenesis of HIV associated encephalopathy is the aim of the work, then the broad results do not allow to shed light on the role of lncRNA in HIV associated encephalopathy.

Thus I suggest that the authors rank their finding , it may be the genes or the pathways , but ranking can certainly help in shedding light on the the role of lncRNAS in disease.
like the reference paper

Zhang C, Wang C, Jia Z, et al. Differentially expressed mRNAs, lncRNAs, and miRNAs with associated co-expression and ceRNA networks in ankylosing spondylitis. Oncotarget. 2017;8(69):113543-113557. doi:10.18632/oncotarget.22708.
give clear target genes for spondylitis. I suggest the authors use this paper as a standard as their works are similar.

Additional comments

Figure 3 can be made more clearer
Some references are cited asnumbers in the text while others as author names. stick to one.
Table 3 with an obscure pvalue is not discussed in text
English language can be checked for errors.
Hive abbreviation used both for disease as well as dataset is confusing sometimes.
Figure 2 says normal as a dataset instead of control

Reviewer 2 ·

Basic reporting

The article "Explorations of the involvement of LncRNA in the occurrence and development of HIV-associated encephalitis using bioinformatics" have using publicly available data for suggest the involvement on LncRNA in the HIV-associated encephalitis.

The presentation in the paper too low on the concept and association with the earlier literature. Though the authors have mentioned that lncRNA have been found to be associated with the pathogenesis of HIV but failed to mention the motive behind the whole work. It can considered that apart from lncRNA there could be several other factors and there is no basis of considering only lncRNA for HIVE until it is proven in that context.

The English is satisfactory though at some place ambiguous. In introduction the author failed to explain the background of the work.

The table are mere mention the list without any detailing in the text which may have helped in some relevant conclusion. The authors failed to connect any specific finding with any results.

Experimental design

Though the research question was well defined but the authors failed explain the methodology clearly. In the methodology section the authors have not mentioned the methodology to find the exact method of their finding. Even though the data is found on the public database but that also require the statistical testing which is not mentioned in the section.

Also GO statistics is not explained to validate the outcome of the experiment.
the data in the public databases are raw and worked out in varied circumstances hence one cannot merely combine data to reach to a general conclusion. Due to high variability and varied platforms one may get high degree of variation which may influence the overall results. Hence the author should have used and mentioned the statistical approach to explain and bring the data to common platform for statistical analysis.

Validity of the findings

Since the methodology and not very elaborated and well explained hence one cannot rely on the finding of the data. In absence of any concrete results, all the finding seems to be too generalized and does not provide any basis for specific finding.

Statistics not explained. Conclusions are too general for any significant findings.
The conclusion provided does not connect to previous findings.

Reviewer 3 ·

Basic reporting

The manuscript is fairly easy to read in the most parts. Some major writing issues are: (1) Multiple times the authors use description such as "differentially expressed probes were screened out" to state that genes represented by certain probes were selected. Their choice of words like "screened out" is confusing as that may imply that probes were left out. (2) The Result section simply lists what is described in figures or simply state the approach once again. The authors do not provide any reasoning/motivation for why a particular analysis was carried out, why did they choose particular datasets and tools etc. Also, there is lot of repetition in Results and Discussion section. For example, the lines 190-215 simply restate what ahs already been said in the results section. This can be clearly seen when results from lines 138-148 are reiterated again in Discussion on lines 197-200 in a very similar manner. As pointed above, the second half the discussion has a clear disconnect with findings of the paper. Discussion should focus more on what the authors think the relevance of their results is in the context of existing knowledge. (3) The heatmaps in Figure 1 are not labeled at all. There is no way to tell what different rows in the heatmaps are representing. What is represented by red and blue colors? What are the scales for the heatmaps? Were the genes shown here clustered in any way?

Experimental design

Overall, the research question is clearly defined, is meaningful and should fill a knowledge gap. The most important issue is that there is no rationale/reasoning provided as to the connection between differentially regulated lncRNAs, miRNAs targeting them, and mRNA targets of these miRNA. The authors should discuss why/how change in expression of lncRNAs may be linked to activity of miRNAs that target lncRNAs and mRNAs. The discussion lists several pathways and processes that are important for neural function and disease but the authors do not discuss how the lncRNAs they identified to be differentially expressed can be affecting these processes. Also, is there a reason why lncRNAs may be more differentially expressed in white matter as compared to frontal cortex or basal ganglia? Is there anything known about heightened lncRNA function in the white matter?

Validity of the findings

Important issues to address: (1) Appropriate statistical evaluation of overlaps between differentially expressed genes in different samples in Figure 1 is missing. (2) How lncRNA expression changes in white matter can be linked to HIVE is not clear.

---

## Round 0.2 · accepted · Accept

Dear Dr. Meiling,

I am pleased to inform that your manuscript entitled "Exploration of the involvement of LncRNA in HIV-associated encephalitis using bioinformatics" has been accepted for the publication after the major revision.

Best regards,

Ravi Tandon, PhD

Reviewer 1 ·

Basic reporting

The article in the present form is self explanatory and much clearer than the earlier forms.
the authors have clearly tried to implement the suggestions provided by this reviewer to their manuscript.The english has been significantly improved and significant literature to related work has been provided.However kindly address these minor queries
=>The HAD in abstract is it same as HAND in the text - please elaborate.
=> -NMDAR- what is this.(Discussion)

Experimental design

the research problem has been well presented and all the questions that arose were nicely answered in the discussion part.The novelty of the paper lies in re annotation of microarray data which is nicely highlighted throughout the paper.Indeed a unique paper.

Validity of the findings

the paper tries to shed light on the pathogenesis of HIVE.the involvement of different nervous system patways indeed corroborates that the overexpression of lncRNAmay be causing multiple pathway to be unregulated , thus causing HIVE.
However ,the authors can also suggest the further research steps to be implemented in the lncrna area

Additional comments

overall a good paper in the present shape.
=>english can be double checked. usage of 'the' is unwanted in some areas
=> future prospective can be included
=> work references of lncrna microarray annonation in other diseases can also be included
eg.,
Guangde Zhang, Haoran Sun, Yawei Zhang, Hengqiang Zhao, Wenjing Fan, Jianfei Li, Yingli Lv, Qiong Song, Jiayao Li, Mingyu Zhang & Hongbo Shi ,(2018)Characterization of dysregulated lncRNA-mRNA network based on ceRNA hypothesis to reveal the occurrence and recurrence of myocardial infarction,
Cell Death Discoveryvolume 4, Article number: 35.